# Construction of Mechanically Reinforced Thermoplastic Polyurethane from Carbon Dioxide-Based Poly(ether carbonate) Polyols via Coordination Cross-Linking

**DOI:** 10.3390/polym13162765

**Published:** 2021-08-17

**Authors:** Gaosheng Gu, Jincheng Dong, Zhongyu Duan, Binyuan Liu

**Affiliations:** Hebei Key Laboratory of Functional Polymer, School of Chemical Engineering and Technology, Hebei University of Technology, Tianjin 300130, China; ggspolymer@126.com (G.G.); jinchengdongpoly@126.com (J.D.); zyduan@hebut.edu.cn (Z.D.)

**Keywords:** thermoplastic polyurethane, poly(propylene ether carbonate) diol, isophorone diisocyanate, in-situ reaction, coordination enhancement

## Abstract

Using carbon dioxide-based poly(propylene ether carbonate) diol (PPCD), isophorone diisocyanate (IPDI), dimethylolbutyric acid (DMBA), ferric chloride (FeCl_3_), and ethylene glycol (EG) as the main raw materials, a novel thermoplastic polyurethane (TPU) is prepared through coordination of FeCl_3_ and DMBA to obtain TPU containing coordination enhancement directly. The Fourier transform infrared spectroscopy, ^1^H NMR, gel permeation chromatography, UV−Vis spectroscopy, tensile testing, dynamic mechanical analysis, X-ray diffraction, differential scanning calorimetry, and thermogravimetric analysis were explored to characterize chemical structures and mechanical properties of as-prepared TPU. With the increasing addition of FeCl_3_, the tensile strength and modulus of TPU increase. Although the elongation at break decreases, it still maintains a high level. Dynamic mechanical analysis shows that the glass-transition temperature moves to a high temperature gradually along with the increasing addition of FeCl_3_. X-ray diffraction results indicate that TPUs reinforced with FeCl_3_ or not are amorphous polymers. That FeCl_3_ coordinates with DMBA first is an effective strategy of getting TPU, which is effective and convenient in the industry without the separation of intermediate products. This work confirms that such Lewis acids as FeCl_3_ can improve and adjust the properties of TPU contenting coordination structures with an in-situ reaction in a low addition amount, which expands their applications in industry and engineering areas.

## 1. Introduction

Thermoplastic polyurethanes (TPUs) are one of the most interesting types of PUs with versatile applications ranging from consumer products, automobile parts, sporting goods, and electronic/medical devices [1,2,3,4,5]. Polyol is an essential component to generate the soft segments in polyurethane (PU) manufacture, which play significant roles in mechanical properties, chemical and oxidative stability for the resulted TPUs [6,7,8]. Carbon dioxide-based polyol can be considered as polyether polyol modified by the introduction of carbonate structure, which is a kind of poly(ether carbonate) (PEC) polyols and combines the advantages of polycarbonate polyol and polyether polyol. Attributing to carbonate units, PU based on carbon dioxide polyol possesses high mechanical properties, high hardness, well wear resistance, and fine solvent resistance [9,10,11]. It is demonstrated that PU obtained from carbon dioxide-based PEC polyols exhibit better hydrolysis resistance and oxidative stability compared with that based on petroleum polyester and polyether type polyols [12,13]. Furthermore, the life cycle assessment studies indicate that the global warming impact of PEC polyols containing 20 wt.% CO_2_ is lower than that of conventional polyether-based polyol by 11–19%, and save fossil resources by 13–16%, possessing environmental and economic benefits [14]. Different from the waxy solid polycarbonate polyol, flexible ether linkages in the backbone enable carbon dioxide-based PEC polyol in a liquid state, allowing facile fabrication of TPUs through typical procedures [15,16]. So carbon dioxide-based PEC polyol is currently attracting great interest as a promisingly alternative feedstock for the preparation of PUs. Carbon dioxide-based polyols with less carbonate unit content usually possess low viscosity which is favorable to use and process but performs worse in mechanical properties [17,18], for which commercial carbon dioxide-based polyols usually possess low carbonate unit. Then, it makes sense to improve the mechanical properties of TPU from carbon dioxide-based polyols so as to expand the application fields.

Metal–ligand coordination interactions are important kinds of noncovalent bonds, and the bond strength can be tuned by changing the combination of metal salt and ligand in a rather broad range [19,20,21]. Thermodynamically stable whilst kinetically labile coordination bonds have been introduced into polymer structures in order to improve the mechanical strength of polymeric materials [22,23,24,25,26]. From the applicable viewpoints, the accessibility of the starting materials and the degree of difficulty in synthesizing the ligand and incorporating the ligand into polymeric materials are important and should be mainly considered. Commercially available dimethylolbutyric acid (DMBA) is a good candidate, in which the –OH groups preferentially react with isocyanate groups [27,28]. Moreover, DMBA has better solubility in different solvents and is regarded as a new generation of green environmental hydrophilic chain extender, which has been widely used in the manufacture of waterborne polyurethanes. Particularly, the carboxylate of DMBA is able to coordinate with metal cations in monodentate, bidentate, or polydentate bindings and various bridging modes [29,30]. Besides, we are concerned with that in the preparation of polymers containing metal-coordination, researchers [22,23,24,25] generally synthesize polymers with coordination structures first, and then dissolve them in organic solvents or water and further coordinate with metal compounds, which is complex and inconvenient in industrial productions and applications. To improve the drawback, we aim at preparing TPU containing metal-coordination with an in-situ reaction and low addition amount of FeCl_3_ to expand its practical applications.

Herein, poly(propylene ether carbonate) diol (PPCD) chosen as a representative carbon dioxide-based PEC polyol, TPUs containing coordination enhancement possesses simultaneously high tensile strength and high toughness are successfully prepared by pre-polymerization through an in-situ method [31,32]. The chemical structures, thermal and mechanical properties of resulted polyurethane are inspected by Fourier transform infra-red (FT−IR) spectroscopy, ^1^H NMR, tensile testing, dynamic mechanical analysis (DMA), and thermogravimetric analysis (TGA).

## 2. Materials and Methods

### 2.1. Materials

PPCD (Industrial Grade) was purchased from Huizhou Dayawan Dazhi Fine Chemical Ltd. (Huizhou, China), whose model is PPCD222 and ^1^H NMR spectrum and detailed data is shown in Appendix A. Isophorone diisocyanate (IPDI, 99%), ethylene glycol (EG, 98%), ferric chloride (FeCl_3_, 98%), and DMBA (99%) were purchased from Heowns Opde Technologies Ltd. (Tianjin, China). Dibutyltin dilaurate (DBTDL, 97.5%) was purchased from J&K Chemical Ltd. (Beijing, China). EG was properly purified by vacuum distillation. PPCD was dehydrated under vacuum at 110 °C before use. DMBA was dehydrated under vacuum at 100 °C. The materials not mentioned were used as received.

### 2.2. Preparation of TPU

#### 2.2.1. Preparation of TPU by One-Step Polymerization Method

PPCD, EG, DMBA, FeCl_3_, and DBTDL were mixed with a few solvents of THF and ether evenly, and then calculated IPDI was added and mixed. The reaction mixture was poured into the mold that had dried and preheated at 90 °C. The mixture was vacuumed until bubbles disappear, reacting at 90–130 °C for 16 h. The as-prepared samples are entitled as 0-DMBA-TPU, 1-DMBA-TPU, 2-DMBA-TPU, and 3-DMBA-TPU, respectively, in which 0-DMBA-TPU excludes DMBA and DMBA content increases gradually by the serial number, whose feeding amount is shown in Table 1.

#### 2.2.2. Preparation of TPU Reinforced with FeCl_3_ by Pre-Polymerization Method

The feeding amount is the same as 2-DMBA-TPU in 2.2.1 except for the amount of FeCl_3_. DMBA dissolved in THF and FeCl_3_ dissolved in ether, and then the two were mixed and stirred for complete reaction. Then the complex of DMBA and FeCl_3_ in the mixed solvent (FeCl_3_−DMBA) was obtained. PPCD, FeCl_3_−DMBA, DBTDL was added into a three-neck flask with an electric stirrer. Then calculated IPDI was added into the flask to react for 3 h at 85 °C under argon atmosphere. Meanwhile, a few THF was added to reduce the viscosity. Then EG was added to the reaction mixture that had cooled to 60 °C to react for another 10 min after the pre-polymerization system stabilized [33]. The reaction mixture was poured into a preheating mold to continue the polymerization to obtain TPU. The mixture was vacuumed until bubbles disappear, reacting at 90–130 °C for 16 h, whose synthesis diagram is shown in Scheme 1 and the model is shown in Figure 1. The as-prepared samples are entitled as 0/18 Fe-TPU, 1/18 Fe-TPU, 2/18 Fe-TPU, and 3/18 Fe-TPU, in which X/Y stands for the mole ratio of *n* (FeCl_3_)/*n* (DMBA).

### 2.3. Analysis and Measurements

FT−IR spectra between 400–4000 cm^−^^1^ were recorded on a Bruker VECTOR-22 spectrometer (Bruker, Karlsruhe, Germany), in which the sample was cut into small pieces to test using KBr pellets. ^1^H NMR spectra were recorded on a Bruker-400 spectrometer (Bruker, Karlsruhe, Germany) at the frequency of 400 MHz. The solvent is DMSO-*d6* or CDCl_3_ and chemical shifts are given in ppm relative to TMS. Gel permeation chromatography tests were performed on an Agilent 1260 Infinity instrument (GPC, Agilent technologies, Waldbornn, Germany), which determined number average molecular weight (*M_n_*) and molecular weight distribution (*Đ*) with a refractive index detector, calibrated with polystyrene standards. The columns used were MIXED-B 10 um 300 × 7.5 mm columns held at 35 °C, using THF as eluents at a flow rate of 1.0 mL/min. UV−Vis spectroscopy was performed on a Cary 60 spectrometer (UV−Vis, Agilent Technologies Ltd., Palo Alto, CA, USA). The samples were dissolved in THF to 10^−3^ mol/L, THF used as the reference solution. The tensile tests were performed on a CMT 1206 electronic testing machine (Nss Laboratory Equipment Ltd., Shenzhen, China). The samples were tested after placing at 25 °C for 24 h, which had dried at 80 °C for 24 h. The type is 3 in ISO 37-2017 and the tensile rate is 100 mm/min. DMA tests were performed on a Tritec 2000 DMA equipment (Triton Technology Ltd., Nottinghamshire, UK), which determined Storage modulus (*E’*) and glass transition temperature (*T_g_*). The tests were carried out with the single cantilever mode at the frequency of 1 Hz and the constant applied strain of 0.1% within −50–60 °C at the heating rate of 3 °C/min under nitrogen atmosphere, whose size of the sample is 40 mm × 7 mm × 2 mm. X-ray diffraction tests were performed on a MiniFlex600 diffractometer (XRD, Rigaku Ltd., Tokyo, Japan). The tests were carried out on the reflection mode at the scanning speed of 4°/min within the range of 5–80° at the step size of 0.02°. Differential scanning calorimetric tests were performed on a Dimond DSC equipment (DSC, PerkinElmer, Waltham, MA, USA) and an empty aluminum crucible was used as the reference. The samples were tested at a rate of 10 °C/min over the range of −50–200 °C in the nitrogen atmosphere after removing the thermal history. TGA tests were recorded on an SDT-Q600 system (TA Instruments, Newcastle, WY, USA) and an empty alumina crucible was used as the reference. The tests were carried out within the range from 40 to 600 °C at a heating rate of 10 °C/min under a nitrogen atmosphere at a flow rate of 100 mL/min.

## 3. Results and Discussion

### 3.1. Chemical Structure of TPU

The chemical structure of TPU is subjected to FT−IR and ^1^H NMR analysis as shown in Figure 2 and Figure 3. FT−IR spectra correspond to PPCD, 0/18 Fe-TPU, 1/18 Fe-TPU, and 3/18 Fe-TPU respectively.

In PPCD, the absorption peak at 1749 cm^−1^ is assigned to stretching vibrations of C=O in the carbonate group. The absorption peaks at 1261 cm^−1^, 1095 cm^−1^ are ascribed to C–O of the carbonate group and C–O of the ether linkage [10,11], which also appears in spectra of TPU. The absorption peak near 3394 cm^−1^ should be assigned to stretching vibrations of N–H, which are located within the hard segment domains. The absorption peak near 1749 cm^−1^ is attributed to C=O of the carbamate group, and the absorption peak at 1531 cm^−1^ is the characteristic absorption peak of C–N–H [34,35,36]. Therefore, the presence of the absorption peaks at 3394 cm^−1^, 1749 cm^−1^, and 1531 cm^−1^ confirmed that TPU is successfully prepared [10].

Meanwhile, ^1^H NMR spectra of PPCD and TPU prepared by pre-polymerization were performed, using DMSO-*d6* as the solvent. Figure 3a–c correspond to PPCD, 0-DMBA-TPU, and 2-DMBA-TPU respectively. 

The assignment of the ^1^H NMR absorption peak is shown in Figure 3 [11,28,34]. The peaks within 4.8–5.2 ppm and 4.0–4.3 ppm in Figure 3a are attributed to CH and CH_2_, which adjoin to the polycarbonate segment. The peaks within 3.2–4.0 ppm are assigned to CH and CH_2_, which connect to the ether linkage. The peaks in PPCD are not subdivided in Figure 3b,c. The peaks at 7.1 ppm and 8.0 ppm are attributed to N–H in the carbamate group. The assigned peaks in Figure 3b,c coincide with ^1^H NMR spectrum of typical PEC-based polyurethane [10,11], indicating that TPU is successfully prepared. The active hydrogen peak of the carboxyl group at 12.8 ppm in 2-DMBA-TPU is found through local amplification between 12.2–13.2 ppm, which is absent in Figure 3a,b and further confirms that the carboxyl groups have been introduced via DMBA chain extender.

### 3.2. Effect of DMBA on Mechanical Properties of TPU

The introduced carboxyl groups can form hydrogen bonds with themselves or carbamate, affecting the performance of TPU [23]. More importantly, the amount of DMBA determines the number of the carboxyl groups that interact with FeCl_3_, so it is important and reasonable to study the effect of DMBA content on the mechanical properties of TPU. Therefore, a series of TPUs with different DMBA content were synthesized by a relatively simple one-step polymerization method at R = 1.05 (R = [–NCO]/[–OH]) [37], and the results are shown in Table 1 and Figure 4.

As shown in Table 1 and Figure 4, the tensile strength of TPU gradually increases with the increasing DMBA content. The tensile strength of 0-DMBA-TPU is 6.5 MPa, which increases to the maximum of 11.7 MPa gradually. The elongation at break of 0-DMBA-TPU is 770%. After the introduction of DMBA, the elongation at break all increase to about 1000%. The larger ethyl side chains of DMBA reduces chain mobility and the chains are hard to align to recrystallize, which leads to better mutual solubility between soft segment and hard segment and then results in higher mechanical properties [27]. To sum up, it shows that DMBA as the chain extender can enhance both strength and toughness to a certain extent. GPC tests show that *M_n_* of TPU increases with the increasing DMBA content. The reason may be that the molecular weight of DMBA is larger than that of EG, whose GPC curves are shown in Appendix A.

### 3.3. Effect of FeCl_3_ on Polymerization Process of TPU

The experimental results show that direct addition of FeCl_3_ will unstabilize the pre-polymerization system and lead to such adverse phenomenon as gelation and difficult molding. It is a valid strategy that FeCl_3_ coordinates with DMBA to minimize the adverse effect. Aiming to study the effect of FeCl_3_ on polymerization process in pre-polymerization method with pre-coordination of FeCl_3_ and DMBA, the tests of UV−Vis, FT−IR spectra, GPC, and ^1^H NMR were used to characterize the coordination effect of FeCl_3_−DMBA, and prepolymer respectively. UV−Vis spectra of DMBA, FeCl_3_, and FeCl_3_−DMBA are shown in Figure 5a. The concentration of DMBA is 10^−3^ mol/L (*n* (FeCl_3_)/*n* (DMBA) = 3/18) and the reference solvent is the mixture of THF and ether (*v*(THF)/*v*(ether) = 1/1). DMBA has almost no absorption and the absorption peaks of FeCl_3_−DMBA are different from that of FeCl_3_ significantly [38]. Furthermore, the DMBA solution is colorless and transparent, the FeCl_3_ solution is yellow-green, and the FeCl_3_−DMBA solution is orange-red, whose color-changing is caused by the coordination effect between the carboxyl groups and FeCl_3_. The results show that there is an obvious coordination effect between DMBA and FeCl_3_.

The prepolymer of 0/18 Fe-TPU (0/18 Fe-TPU-Prepolymer) and 3/18 Fe-TPU (3/18 Fe-TPU-Prepolymer) were simultaneously sampled for tests of FT−IR spectra, GPC, and ^1^H NMR to study the effect of FeCl_3_ on the prepolymer, and the result are shown in Figure 5b,c and Appendix A respectively.

As shown in Figure 5b, FT−IR spectra of 0/18 Fe-TPU-Prepolymer and 3/18 Fe-TPU-Prepolymer coincide, which are similar to TPU in Figure 2 except for the intense –NCO absorption peak. The similar phenomena happen in the testing of GPC and ^1^H NMR, in which the testing results of 0/18 Fe-TPU-Prepolymer and 3/18 Fe-TPU-Prepolymer are similar in Figure 5c and Appendix A respectively. The results confirm that the introduction of FeCl_3_ has no significant effect on prepolymer and then the polymerization process. Figure 5d shows the physical pictures of TPU, in which the color of the sample without FeCl_3_ is transparent and gradually deepens to dark brown with the increasing amount of FeCl_3_.

### 3.4. Mechanical Properties of TPU Reinforced with FeCl_3_

The mechanical properties of TPU reinforced with FeCl_3_ are shown in Figure 6 and Table 2, whose addition amounts of FeCl_3_ correspond to 0.0 wt.%, 0.3 wt.%, 0.5 wt.%, 0.8 wt.%, and 1.0 wt.% from Entry 1 to Entry 5 respectively. Generally, DMBA in Table 2 is potential to coordinate with more FeCl_3_. Because the viscosity of the reaction system gradually increases with the increasing addition of FeCl_3_, the processing becomes difficult, for which the value of *n* (FeCl_3_)/*n* (DMBA) up to 4/18 to the maximum. The tensile strength and modulus gradually increase with the increasing addition of FeCl_3_. The tensile strength of 0/18 Fe-TPU is 10.3 MPa, which increases to the maximum of 17.8 MPa with an increasing rate of 72.8%. The tensile modulus also increases gradually with the increasing addition of FeCl_3_, whose value increases from 14.8 MPa to the maximum of 93.9 MPa. On the other hand, the elongation at break decreases as the addition amount of FeCl_3_ increases. The reason for this phenomenon is that the density of crosslinks constructed by the carboxyl groups and FeCl_3_ increase, thus resulting in higher tensile strength, modulus, but lower elongation at break [23,25].

### 3.5. DMA Tests of TPU Reinforced with FeCl_3_

To study *E’* and *T_g_* of TPU reinforced by FeCl_3_, DMA tests were conducted, and the results are shown in Figure 7. It is obvious that *E’* of TPU gradually increases with the increasing addition of FeCl_3_, which reflects that the motility of the chains in TPU gradually decreases with increasing density of coordination cross-linking. The restricted chain mobility means the chains are more rigid and less easy to move, which results in a higher *E’* [24,39].

*T_g_* determined by DMA gradually moves to high temperature with the increasing addition of FeCl_3_. *T_g_* of 0/18 Fe-TPU is 27.4 °C, and that of 4/18 Fe-TPU rises to the maximum of 52.5 °C. The reason is that the density of coordination cross-linking increases, which restricts the motility of the TPU chains, so *T_g_* gradually moves toward a high temperature with the increasing addition of FeCl_3_.

### 3.6. Crystallization Performance of TPU Reinforced with FeCl_3_

To study the crystallization performance of TPU reinforced by FeCl_3_, XRD tests of Entry 1–Entry 5 in Table 2 were performed and the results were shown in Figure 8. It is obvious that the spectra appear typical diffuse scattering peak at 2*θ* = 19°, which means the crystallinity of relatively ordered hard segments domains in polymers [5,40]. Then, it confirms that the TPU reinforced by FeCl_3_ or not are amorphous polymers, and also it is mainly the reason why the melting peak of TPU is not observed in DSC tests in Appendix A.

### 3.7. Thermal Stability of TPU Reinforced with FeCl_3_

To study the thermal stability of TPU reinforced with FeCl_3_, TGA tests Entry 1–Entry 5 in Table 2 were conducted and the results are shown in Figure 9 and Table 3. *T*_5%_ gradually decreases with the increasing addition of FeCl_3_, which is usually used as the onset decomposition temperature [27]. *T*_5%_ of 0/18 Fe-TPU is 266.4 °C and *T*_5%_ of 4/18 Fe-TPU decreases to 243.8 °C in Figure 9a and Table 3. *T*_10%_ and *T_max_* also decrease similarly, demonstrating the thermal stability of TPU decreases with the introduction of FeCl_3_. On the one hand, the coordination bonds rupture at a high temperature and the ferric compound evenly distributes in TPU, which is equivalent to a physical filler that weakens the interaction, resulting in the decrease in thermal stability [41]. On the other hand, many researchers have widely revealed that the interaction of such Lewis acid as transition metals in the polymer can boost thermal degradation of the organic matrix, leading to adverse effects [42,43,44].

As shown in Figure 9b. Two thermal decomposition peaks can observe through DTG curves. The peak within 260–290 °C corresponds to the hard segment, and the other within 290–320 °C corresponds to the soft segment [11,27]. Both gradually move to lower temperatures with the increasing addition of FeCl_3_ for the same reason as *T_g_* curves.

## 4. Conclusions

In summary, carbon dioxide PEC polyols-based TPU reinforced with FeCl_3_ was successfully prepared, which was tested and characterized using different approaches, including FT−IR spectroscopy, ^1^H-NMR, GPC, UV−Vis, tensile testing, XRD, DMA, and TGA. By exploring the adding method of FeCl_3_, it is found that TPU prepared through FeCl_3_−DMBA possesses well-comprehensive properties. With the increasing addition of FeCl_3_, the tensile strength and modulus of TPU increased, but the elongation at break decreased. DMA tests show that the *T_g_* moves to a high temperature gradually along with an increasing addition of FeCl_3_. XRD tests indicate that TPUs reinforced with FeCl_3_ or not are amorphous polymers. TGA tests show that thermal stability decreases gradually with increasing addition of FeCl_3_. This synthesis strategy has the advantages of low iron addition, in-situ reaction, and no separation of intermediate products. Compared with the traditional route of first obtaining polymer and then dissolving to coordinate with metal centers, this work provides a more effective strategy for large-scale industrial applications, that is, adding FeCl_3_ in the preparation stage and directly obtaining TPU with coordination enhancement.

## Data Availability

The data presented in this study are available.

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
