# Peer review of "Construction of Mechanically Reinforced Thermoplastic Polyurethane from Carbon Dioxide-Based Poly(ether carbonate) Polyols via Coordination Cross-Linking"

_polymers, 2021, doi:10.3390/polym13162765_

Round 1

Reviewer 1 Report

  1. The manuscript is well organized, and the subject is well presented. The methods used are sound and the presentation and discussion of results is logical.
    The manuscript requires some major revisions to bring it to a level worthy of publication. My recommendations are detailed below:
  2. The current study investigates the fabrication of thermoplastic polyurethane (TPU) and study its characteristics and mechanical performance. For this, the authors test the tensile strength, DMA and other non-destructive techniques. The authors found that increasing the content of ferric chloride improves the mechanical properties of the material. While other effects on DMA and microstructure of the polymer were observed.
  3. The abstract reads well but please consider the following: Please consider reviewing the abstract and highlight the novelty, major findings and conclusions.
  4. Please remove the word complex in line 13, please avoid using excessive wording in any sentences which do not add any value to the sentence.
  5. Please move Figure 1 to below the materials and methods section.
  6. Introduction is short and can be improved/expanded, please discuss on past studies similar or closely related to this work, report what they did and what were their main findings/differences/similarities and explain how does your study brings new knowledge and difference to the field.
  7. Before the last paragraph in the introduction, the authors should answer the following question: What is the research gap did you find from the previous researchers in your field? Mention it properly. It will improve the strength of the article.

  8. Line 154-155 “The change of absorption…..” is this a speculation or a fact, in either way please support with references and compare with past studies.
  9. Please move figure 3 to after line 173.
  10. Please add figures after the first mentioned in the manuscript and not before. Please check for this issue elsewhere.
  11. For Table 1, why the mechanical properties are not shown with a +/-, this make me raise a question on how many times each sample was tested for? Ideally it should be at least five repetitions and then there should be an average reported with an error range. Please explain. I am saying this because you are showing some error range in Figure 4b but it is now seen in the table!
  12. Elongation at break is >1000% is that correct?
  13. The quality of images and graphs in figure 5 can be improved.
  14. Line 232 “The Mechanical property” properties and not property.
  15. Conclusion is weak and must be improved, the authors are encouraged to use bullet points and list most important findings from each of the results and discussion sections.
  16. Overall clear and well presented study.

Reviewer 2 Report

Dear authors,

The article “Construction of Mechanically Reinforced Thermoplastic Polyurethane from Carbon Dioxide-Based Poly(ether carbonate) Polyols via Coordination Cross-linking”

The work presents interesting results. However, it needs to be better explored. Although the justification that there is a reduction in carbon dioxide is presented, this is not the focus of the data presented. What is being proposed is the formation of bonds between the metallic site and the polymer.

So the introduction needs to be better focused and presented what matters with the article. Some terms should be corrected, and an example is on page 2, line 60. The polymerization reaction of urethane does not generate by-products, so it is usual to use a step reaction and not polycondensation.

The figure shown in scheme 1 shows the possible interaction between the metal and the carbonyl group. However, all authors' samples contain DMBA. Therefore, there is a need to include a control sample to verify that this type of bond only forms when the DMBA is present.

Experimental.

In molecular mass measurements, using THF and not 0.1% BrLi DMF will affect the molar mass values, as there is a considerable amount of hydrogen bonding in the system.

Results

In the region of the carbonyl band (~1749) a more intense splitting of the band with the amount of metal is noticed. To what fact is this attributed? The formation of carboxylate groups in the head does not only occur through the addition of DMBA. Note that the relative density of hydrogen bonds will change, as will the degree of phase separation. It needs to be further explored. See at: 10.1002/app.50709

A point that caught my attention is regarding the dynamic mechanical properties. In which the Y axis of Figure 7 should be presented in a logarithmic scale. Another point is that there is no clarity about the modulus value at room temperature, which could be helpful because the authors could use the theory of elasticity and estimate the crosslink increase, which is mentioned in the paper's title. What is the degree of crosslink on the TPUs? Or is this partial? What level?

The glass transitions of the flexible and rigid phases were detected by DSC and not by DMA. This seems inconsistent, as this technique is much more accurate for this purpose. What attentive evidence is there that the transition shown in the supplementary files is in fact, the Tg of the rigid phase? Is there some theoretical calculation, or even pure rigid phase measurement, that could help?

However, this is amorphous polymer diffraction. What is the meaning of the XRD peak? What is the average distance between chains? 10.1016/j.porgcoat.2017.04.002, 10.1002/app.38196

Round 2

Reviewer 1 Report

All questions answered and paper can be accepted

Reviewer 2 Report

The work can be accepted. Although the title is inappropriate for the scope, as it does not have the crosslink degree.